# Modeling the Batch Sedimentation of Calcium Carbonate Particles in Laboratory Experiments—A Systematic Approach

**DOI:** 10.3390/ma16134822

**Published:** 2023-07-04

**Authors:** Maria J. Moura, Carolina S. Vertis, Vítor Redondo, Nuno M. C. Oliveira, Belmiro P. M. Duarte

**Affiliations:** 1Instituto Politécnico de Coimbra, Instituto Superior de Engenharia de Coimbra, Rua Pedro Nunes, Quinta da Nora, 3030-199 Coimbra, Portugal; bduarte@isec.pt; 2Centro de Investigação em Engenharia dos Processos Químicos e dos Produtos da Floresta, Universidade de Coimbra, Rua Sílvio Lima, Pólo II, 3030-790 Coimbra, Portugal; carol@eq.uc.pt (C.S.V.); nuno@eq.uc.pt (N.M.C.O.); 3LED&MAT, Instituto Pedro Nunes, Rua Pedro Nunes, Edifício A, 3030-199 Coimbra, Portugal; vitor@ipn.pt

**Keywords:** batch sedimentation, velocity of settling, model fitting, particle aggregation, calcium carbonate

## Abstract

The design of continuous thickeners and clarifiers is commonly based on the solid flux theory. Batch sedimentation experiments conducted with solid concentrations still provide useful information for their application. The construction of models for the velocity of settling allows the estimation of the flux of solids throughout time, which can, in turn, be used to find the area of the units required to achieve a given solid concentration in the clarified stream. This paper addresses the numerical treatment of data obtained from batch sedimentation experiments of calcium carbonate particles. We propose a systematic framework to fit a model that is capable of representing the process features that involve (i) the numerical differentiation of data to generate initial estimates for the instantaneous velocity of settling; (ii) the integration of a differential equation to fit the model for the velocity of settling; and (iii) the assessment of the quality of the fit using common statistical indicators. The model used for demonstration has a theoretical basis combined with an empirical component to account for the effect of the particle concentrations and their state of aggregation. The values of the numerical parameters obtained are related to the characteristic dimensions of the aggregates and their mass-length fractal dimensions.

## 1. Motivation

The analysis of batch settling velocities provides useful information for the design of continuous gravity settlers, for the separation of particulate solids from a slurry. This practice has its roots in the classic development of sedimentation theory and continues to be used because batch settling data are easy to obtain from laboratory experiments and they provide lower bounds for sizing [1] (Chapter 5). The measurement of the height of the interface between the diluted and concentrated phases formed in controlled experiments in a glass cylinder allows for estimating the velocity of settling, which, in turn, can be used to determine the solid flux and the area necessary to safely operate the unit [2]. Typically, very fine particles settle slowly, while if aggregation occurs, larger velocities are expected. In practice, many factors that depend on the (i) particle properties; and (ii) flow characteristics influence the flow dynamics and the velocity of settling. Apart from those factors, the assumption that a particle settles freely without interference from other particles is inadequate for high-concentration suspensions. Often, the term free settling is applied to systems where the particles are able to settle individually, and hindered settling (or thickening) is used to describe the behavior at high particle concentrations. For many systems of practical interest, the sedimentation rates depend on the concentration and the state of aggregation of the particles, rather than the particle size alone [3]. The incorporation of aggregation in steady-state models describing the velocity of settling in a thickener was considered by Usher et al. [4] and later improved by Zhang et al. [5]. Our study focuses on the region of low solid concentrations of micrometer-sized calcium carbonate particles with irregular shapes, where the occurrence of aggregation is required to explain the behavior of the settling velocities observed experimentally.

After collecting their experimental data, several authors use least squares or weighted least squares regression to fit the height of the interface to a given model; see Torfs et al. [6] for an example. The fitted models range from purely empirical forms [7,8] to phenomenological-based forms [9,10]. Depending on their structure, the models can be complex to fit, since the height of the interface is often represented by one or more nonlinear ordinary differential equations (ODEs) or partial differential equations. A strategy to circumvent this difficulty consists in using numerical differentiation techniques to determine the instantaneous velocities of settling from consecutive experimental heights of the interface. This simplifies adjusting the velocities since the optimization problem is then described by nonlinear algebraic equations, thus avoiding the numerical integration of the ODEs. However, this numerical differentiation usually also introduces errors in the estimates of the velocity since the measurement errors are also differentiated; original data uncertainty can be drastically amplified by calculating the time derivatives of the data in the absence of adequate smoothing/regularization methods. This point is worthy of attention, and a numerical approach combining the advantages of both strategies might reduce the numerical issues associated with the model fitting while improving its accuracy. Practically, a first estimate of the velocity of settling obtained through the use of finite differences is easy to achieve. This might provide useful initial estimates for the model parameters, which can be later estimated more reliably through the direct integration of the model’s ODEs. Furthermore, in the case of constrained least squares, the efficient estimation of the local derivatives of the differential model can be included in the optimization problem as well, making this procedure more efficient [11].

Another numerical aspect requiring further analysis is related to the assessment of the quality of the model [12]. Quite often, the parameters obtained are highly correlated, and this may lead to uncertain estimates of the velocity of settling when the model is used for prediction [13]. To characterize the confidence bounds of the values predicted by these models, the correct use of the confidence limits determined for the parameters is also crucial. However, much of the literature involving model validation of batch sedimentation data disregards these numerical issues. This paper intends to help fulfill this gap through the use of a systematic approach for treating the regressed data obtained from batch sedimentation experiments. The approach and the tools proposed herein are general concerning their application and can be applied to experiments with larger data volumes as well as to other systems, such as those involving sludge sedimentation.

This paper contains four elements of novelty: (i) A model including the particle aggregation is used to describe the batch sedimentation of calcium carbonate particles in glass cylinders. The model stands on phenomenological principles but is aligned with the empirical representation of Richardson and Zaki [14] for the sedimentation of concentrated suspensions. (ii) The aggregation of particles is explicitly included in the model using fractal metrics. (iii) A numerical approach based on the combination of differential and integral methods commonly used to fit kinetic data is proposed. The velocity of settling is firstly fitted to find initial parameter estimates, and subsequently used to fit the height of the interface. (iv) Confidence intervals are constructed for the parameters and the quality of the fitting is assessed using common performance metrics, such as the parametric cross-correlation and the visual analysis of 95% confidence ellipsoids for pairs of parameters.

The paper is organized as follows. Section 2 introduces the model used to describe batch sedimentation in glass cylinders. Section 3 describes the numerical approach used for model fitting. First, the architecture is presented; next, each step is detailed. In Section 4, we characterize the materials involved in the experiments and the experimental procedure. Section 5 presents the results obtained for a battery of tests carried out at different solid concentrations. Section 6 provides a final overview of the work and a summary of the results.

## 2. Particles Settling Model

In our notation, boldface lowercase letters represent vectors, boldface capital letters stand for continuous domains, blackboard bold capital letters are used to denote the discrete domains, and capital letters represent matrices. Finite sets containing ι elements are compactly represented by 〚ι〛≡{1,⋯,ι}. The transpose operation of a matrix or vector is represented by “${}^{⊺}$”.

First, we introduce the theoretical model for the sedimentation of uniformly sized particles in a column of liquid. Next, the model is extended to non-uniformly sized particles of different shapes. Let ϑ be the velocity vector field of the particles, assumed to have a characteristic diameter $d_{p}$, falling through a Newtonian fluid under gravity. Their density is designated as $\rho_{p}$, the density of the liquid is $\rho_{l}$, and its viscosity $\mu_{l}$. The momentum balance representing the forces interacting with the particle is given by Navier–Stokes (see Equation (Equation 1)) [15]
(1)$$\rho_{l}\;\frac{\partial \vartheta }{\partial t}+\rho_{l}\;\vartheta \;\cdot \;\nabla \vartheta =-\nabla P+\mu_{l}\;\nabla^{2}\vartheta +\rho_{l}\;g$$
where *t* stands for time, *P* for pressure, g for the vector field representing the gravity force, ∇ for the gradient operator, and $\nabla^{2}$ for the Laplacian. The first term on the left-hand side of (Equation 1) represents the moment accumulation per unit volume, the second is for the momentum loss by convection. The first term on the right-hand side represents the loss of momentum due to the pressure, the second the viscous force, and the third the gravitational force. At a steady state, the accumulation term is null. At low flow rates, corresponding to small Re(≪1) numbers, the term $\rho_{l}\;\vartheta \;\cdot \;\nabla \vartheta \ll \mu_{l}\;\nabla^{2}\vartheta$ and Equation (Equation 1) reduces to:(2)$$\mu_{l}\;\nabla^{2}\vartheta =\nabla P-\rho_{l}\;g$$

When the inertia and viscous forces are in equilibrium, the net acceleration of the particles is zero and their velocity (in the *y*-direction) becomes constant. This implies that the viscous force also becomes constant and the bodies continue to move in this direction with constant or terminal velocity, commonly designated as the Stokes velocity, $v_{\mathrm{St}}$ (see Equation (Equation 3)):(3)$$v_{\mathrm{St}}=\frac{\left(\rho_{p}-\rho_{l}\right)\;g\;d_{p}^{2}}{18\;\mu_{l}}$$In opposition, when the accumulation term in Equation (Equation 1) is considered, the dynamics of the velocity of the sphere, v(t), is obtained through the solution of the equation [16]
$$\frac{dv}{dt}+\frac{12\;\mu_{l}}{\rho_{p}\;d_{p}^{2}}v-\frac{2\;(\rho_{p}-\rho_{l})}{3\;\rho_{p}}\;g=0,\;v\left(0\right)=0,$$for which the analytical solution is
$$v\left(t\right)=\frac{\left(\rho_{p}-\rho_{l}\right)\;g\;d_{p}^{2}}{18\;\mu_{l}}\;\left[1-\exp\left(-\frac{\mu_{l}}{27\;\rho_{p}\;d_{p}^{2}}\;t\right)\right].$$

As an illustration, when the particles are of micrometer size and the fluid is not significantly viscous, starting from a state of rest, the particles are able to achieve 99% of $v_{\mathrm{St}}$ in the interval of about 1 ms. This implies that the characteristics associated with the terminal velocity will be more important for the overall description of the system. Moreover, different behaviors are observed for multiple spheres falling in a liquid-filled tube due to their possible interactions with each other. These aspects, together with the interactions with the walls of the tube used, are not considered in the above model.

Happel and Brenner [17] analyzed the sedimentation of multiple spheres in a fluid, assuming that each falls in a spherical unit cell of liquid. The steady-state solution of Equation (Equation 2) for this conceptual framework has the form
(4)$$v_{t}=v_{\mathrm{St}}\;f\left(\epsilon \right)$$
where $v_{t}$ is the terminal velocity of a single sphere in the population, also known as the settling velocity, ϵ is the volumetric fraction of fluid (or media porosity), and f(•) expresses a functional relation. This study also evidences that the settling velocities decrease with the concentration of suspensions, emphasizing the dependence between $v_{t}$ and the concentration of solids. A theoretical relationship between the velocity of settling and the concentration of suspensions was initially proposed by Maude and Whitmore [16]. For an overview of the different functions f(•), the reader is referred to Davies [18]; the most commonly used was developed by Richardson and Zaki [14], who proposed $f\left(\epsilon \right)=\epsilon^{n}$, with *n* being a constant depending on the particles’ Reynolds number. For systems where Re<0.2, *n* was found to be equal to 4.65 [19].

The batch sedimentation experiments considered were performed in graduated cylinders (Figure 1a). Initially (at time $t_{0}$), a perfectly mixed suspension with a concentration of solids $c_{0}=c\left(0\right)$ (here, expressed in g/L and corresponding to a volumetric concentration of $c_{0}/\rho_{p}$) was added to a cylinder provided with a convenient height scale. The initial height of the suspension, $h\left(t_{0}\right)=h_{0}$, was recorded. Then, a chronometer was switched on, and the reading of the height of the interface between the settling particles and the supernatant liquid, designated as $h_{i}=h\left(t_{i}\right)$, at a given set of time instants, $t_{i}$, was carried out; see the scheme in Figure 1b. The test was extended until no relevant change was seen in the height of the interface. We designate this height by $h_{\infty }$ and the corresponding time instant is $t_{\infty }$.

Above the interface, a clarified zone is formed where the concentration of solids is zero. The concentrated zone (shortened to *Concentr. zone* in Figure 1b) is also assumed to remain perfectly mixed, and a single characteristic diameter $d_{p}$ was previously considered, meaning that all particles settle at the same rate. In this case, a steady-state balance of the solids enables the characterization of their respective concentrations throughout the duration of the experiment [3]:(5)$$c\left(t\right)=\frac{c(0)\;h(0)}{h(t)}$$

The porosity of the uniformly mixed suspension in the concentrated zone is, thus, also dependent on time, and is given by:(6)$$\epsilon \left(t\right)=1-\frac{c(t)}{\rho_{p}}=1-\frac{c(0)}{\rho_{p}}\frac{h(0)}{h(t)}$$Therefore, monitoring the position of the interface allows for estimating the instantaneous hindered settling velocity $v_{t}\left(t\right)$
(7)$$v_{t}\left(t\right)=\frac{dh(t)}{dt}=v_{\mathrm{St}}\;\epsilon {\left(t\right)}^{n}=v_{\mathrm{St}}\;{\left(1-\frac{c(0)}{\rho_{p}}\frac{h(0)}{h(t)}\right)}^{n},$$
where the empirical relation of Richardson and Zaki [14] is used to account for the effect of the particle concentration. The parameter *n* can be estimated by fitting v(t) from h(t). Clearly, even after assuming a single characteristic diameter $d_{p}$, the possible effects of the particle shapes when irregular particles are used and the occurrence of particle aggregation are not included in this model.

To account for the agglomeration of particles, we consider that the characteristic dimensions of the aggregates are their fractal dimensions, as in Meakin [20]. A common cause for aggregation is the presence of electric charges in the particles, resulting in the inclusion of ions, water, and other smaller particles in the aggregates that form through coagulation. Let $d_{\mathrm{agg}}$ denote the diameter of the aggregates and $D_{f}$ denote its mass-length fractal dimension [21,22]. The effect of the formation of aggregates on the effective particle diameter in Equation (Equation 7) is explicitly introduced in the model using a new parameter, *k*. An additional parameter η is also needed to denote the fact that the final composition in the concentrated zone also includes a significant fraction of water; this means that the final value of $\epsilon (t_{\infty })$ is different from zero when the settling velocity reaches zero. In this analysis, the value of *n* in Equation (Equation 7) was kept fixed at 4.65, so that the remaining parameter values obtained in this study can be more directly compared with reference values determined using the relation postulated by Richardson and Zaki [19]. The resulting model is
(8)$$v_{t}\left(t\right)=\frac{dh(t)}{dt}=k\;v_{\mathrm{St}}\;{\left(1-\eta \;\frac{c(0)}{\rho_{p}}\frac{h(0)}{h(t)}\right)}^{4.65}$$
which coincidentally has a form similar to the model proposed by Michaels and Bolger [23]. It is worth mentioning that *k* and η are, respectively, equal to non-dimensional numbers ${\left(d_{\mathrm{agg}}/d_{p}\right)}^{D_{f}-1}$ and ${\left(d_{\mathrm{agg}}/d_{p}\right)}^{3-D_{f}}$, considered in Nieto et al. [22] (Equation (Equation 2)).

For compactness, let $\theta ={\left(k,\eta \right)}^{⊺}$ be the vector containing the parameters to be fitted. Each of them is constrained to a compact domain, i.e., $\theta_{i}\in \left[\theta_{i}^{L},\theta_{i}^{U}\right]$, where $\theta_{i}^{L}$ is the lower bound for parameter *i*, and $\theta_{i}^{U}$ is the upper bound. $n_{\theta }$ is the number of parameters to fit in Equation (Equation 8), which in this context is 2.

## 3. Numerical Strategy for Model Fitting

In this section, we present the strategy used to fit Equation (Equation 7) to data resulting from laboratory batch experiments. Similar to the analysis of chemical kinetics rate laws, kinetic data can be analyzed employing the integral method [24,25] combined with robust numerical optimization procedures [26]. This method fits the kinetic rates expressed as a set of differential-algebraic equations to cumulative data (i.e., the reaction extent) calculated by sampling the reaction system during the experiment at a given set of time instants. This approach can be numerically challenging, as it requires a numerical ODE solver combined with an optimizer that handles the nonlinear least squares problem; an adequate initialization of the parameters is crucial to achieving the convergence of the optimizer and increasing the computational efficiency. Apart from this issue, this technique is rather appealing as it allows fitting the data as it was recorded in a bias-free adjustment (i.e., without any previous treatment)—see Schittkowski [27], Edsberg and Wedin [28] for applications.

Alternatively, one may use the differential method, which fits the kinetic rates to increments in each reaction advance [29]. Here, the model to be fitted is represented by algebraic rate equations, which can be linear or nonlinear, and naturally become easier to handle within the least squares procedure than the corresponding model in the integral method. However, the numerical differentiations of the sampled data are required prior to the fitting task. This operation is numerically unstable and amplifies the error of the derivatives with respect to the measurement errors of the acquired signals [30]. This approach was considered by Cremers and Hübler [31], among others. Recently, Vertis [32] proposed a scheme, combining both methods in the context of reaction network identification. The strategy inspired us to systematize the architecture of the model fitting tool used in this study, which appears in Figure 2.

Now, we briefly analyze each step of the algorithm in Figure 2 and their connections; the next sections present an in-depth analysis of the tools involved. Let us consider that the data obtained from the experiments consist of records including the time instants at which the system was observed and the corresponding heights of the interface, i.e., $D^{n_{o}}=\left\{t_{i},h\left(t_{i}\right),\;i\in 〚n_{o}〛\right\}$, $n_{o}$ being the number of data points. These data are used to estimate the instantaneous hindered velocities using Lagrange interpolation (Step 2). The extended data set containing the original data and the estimates of the velocity of settling, i.e., $D^{n_{o},\exp}=\left\{t_{i},h\left(t_{i}\right),v_{t}\left(t_{i}\right),\;i\in 〚n_{o}〛\right\}$, are then used to create initial estimates of the model parameters with the differential method (see Step 3). Here, we use the least squares method to fit the algebraic form of the model and obtain starting parameter estimates, which are fed to Step 4, where the integral method is applied. Least squares is also used in Step 4 to regress the differential equation representing the model, and produce final estimates. The information produced here (i.e., the parameter estimates, the parametric covariance matrix, called $S(\theta^{\mathrm{est}})$, and the local prediction errors given by $e=h-h^{\mathrm{est}}$) is fed to Step 5, where additional model accuracy statistics and metrics for evaluating the quality of fit are determined.

Section 3.1 describes the technique used for constructing the estimates of the velocity of settling using Lagrange polynomial approximations. Section 3.2 states the mathematical formulations of the optimization problems solved to fit the data. Finally, Section 3.3 introduces the metrics used to analyze the quality of fit and the model’s predictive accuracy.

### 3.1. Numerical Differentiation—Step 2

This section describes the approach used for estimating the velocity of settling at sampling time instants $t_{i}$. For this purpose, we use Lagrange polynomials as they can easily cope with data measured at unevenly spaced points. Given an (ordered) set of k+1 time instants represented by $\tau =\{t_{1},\cdots ,t_{k+1}\}$, all distinct, at which the system was sampled, the $k^{\mathrm{th}}$ degree Lagrange polynomial approximation for the height of the interface at $t\in [t_{1},t_{k+1}]$ is as follows:(9)$$h\left(t\right)=\sum_{i=1}^{k+1}h\left(t_{i}\right)\;\prod_{\begin{array}{c} j=1\\ j\neq i \end{array}}^{k+1}\frac{t-t_{j}}{t_{i}-t_{j}}$$Then, we derive (Equation 9) with respect to *t* and construct local estimates for the velocity of the interface. The approximation produced is [33]:(10)$$\frac{dh(t)}{dt}=\sum_{i=1}^{k+1}h\left(t_{i}\right)\;\sum_{\begin{array}{c} j=1\\ j\neq i \end{array}}^{k+1}\left(\frac{1}{t_{i}-t_{j}}\;\prod_{\begin{array}{c} \ell =1\\ \ell \neq (i\vee j) \end{array}}^{k+1}\frac{t-t_{\ell }}{t_{i}-t_{\ell }}\right)$$

In our context, all the instants *t* of interest (those at which dh(t)/dt is estimated) belong to τ; consequently, $t\in \{t_{1},\cdots ,t_{k+1}\}$. To minimize the approximation error, we assume that *t* coincides with the middle point of the series of k+1 points, corresponding to $t_{\lfloor (k+1)/2\rfloor +1}$; ⌊•⌋ is the floor operator. For the first and the last ⌊(k+1)/2⌋ points of $D^{n_{o}}$, *t* coincides with the point closer to the center of $\left[t_{1},t_{k+1}\right]$. Equation (Equation 10) can be used with unequal time intervals $t_{i}-t_{i-1}$.

In our application, we use Lagrange polynomials of the sixth degree. This means that in Equations (Equation 9) and (Equation 10), we have k=6, the derivative in each point includes information of k+1=7 contiguous points, and the middle point (the point at which the derivative is calculated) is the fourth of each set $\left\{t_{1},\cdots ,t_{7}\right\}$.

### 3.2. Model Fitting—Steps 3 and 4

Here, we fit the models for the velocity of settling (Step 3 of the algorithm) and the height of the interface (in Step 4). Model fitting is an LS problem, where the sum of squared errors of the predictions is minimized. Nonlinear programming problems describing the model parametrization in Steps 3 and 4 are, respectively,
(11a)$$\begin{array}{cc} \min_{\theta }\; & \;\sum_{i=1}^{n_{o}}{\left(v_{t}\left(t_{i}\right)-v_{t}^{\mathrm{est}}\left(t_{i}\right)]\right)}^{2} \end{array}$$
(11b)$$\begin{array}{cc} s.t.\; & \;v_{t}^{\mathrm{est}}\left(t_{i}\right)=k\;v_{\mathrm{St}}\;{\left(1-\eta \;\frac{c(0)\;h(0)}{h\left(t_{i}\right)\;\rho_{p}}\right)}^{4.65} \end{array}$$
(11c)$$\begin{array}{cc} \; & \;\theta \in [\theta^{L},\theta^{U}], \end{array}$$
and
(12a)$$\begin{array}{cc} \min_{\theta }\; & \;\sum_{i=1}^{n_{o}}{\left(h\left(t_{i}\right)-h^{\mathrm{est}}\left(t_{i}\right)]\right)}^{2} \end{array}$$
(12b)$$\begin{array}{cc} s.t.\; & \;\frac{dh^{\mathrm{est}}\left(t_{i}\right)}{dt}=k\;v_{\mathrm{St}}\;{\left(1-\eta \;\frac{c(0)\;h(0)}{h^{\mathrm{est}}\left(t_{i}\right)\;\rho_{p}}\right)}^{4.65} \end{array}$$
(12c)$$\begin{array}{cc} \; & \;h^{\mathrm{est}}\left(0\right)=h\left(0\right) \end{array}$$
(12d)$$\begin{array}{cc} \; & \;\theta \in [\theta^{L},\theta^{U}]. \end{array}$$

Equations ([Disp-formula FD11a-materials-16-04822]) and ([Disp-formula FD12a-materials-16-04822]) are the objective functions in the LS procedure, ([Disp-formula FD11b-materials-16-04822]) and ([Disp-formula FD12b-materials-16-04822]) are the equalities representing the model, ([Disp-formula FD11c-materials-16-04822]) and ([Disp-formula FD12d-materials-16-04822]) are the boundedness conditions of the parameters and ([Disp-formula FD12c-materials-16-04822]) is the initial condition for the ODE representing the height of the interface. The problems are solved with the Levenberg–Marquardt algorithm; the details are in Nielsen and Madsen [34]. The problem (12) requires an ODE solver, and we use an algorithm of variable order and variable steps based on the numerical backward differentiation formulae. The numerical tolerance for the optimization was set to 1 × 10^−5^. Practically, to handle problem (12), we use a *sequential* approach, which sequentially iterates between the model solution and the likelihood minimization until convergence [35].

### 3.3. Quality of Fit Metrics—Step 5

We now present the performance indicators used to evaluate the quality of the fitting (see Step 5 of the algorithm).

To measure the predictive accuracy of the model, i.e., the ability to reproduce the data, we use the adjusted *R*-squared correlation coefficient, designated here as $R_{\mathrm{adj}}^{2}$, as proposed by Seber and Wild [36]. To analyze the collinearity of the parameters, we use the covariance matrix $S(\theta^{\mathrm{est}})$ (given by the linearized model from the LS algorithm at convergence) to construct an approximation to the parametric correlation matrix. Let $B(\theta^{\mathrm{est}})$ be a diagonal (square) matrix of size $n_{\theta }\left(=2\right)$ containing the square roots of the diagonal elements of $S(\theta^{\mathrm{est}})$, i.e., $B_{i,i}=\sqrt{S_{i,i}},\;i\in 〚n_{\theta }〛$ and $B_{i,j}=0,\;i,j\in 〚n_{\theta }〛,\;i\neq j$. The correlation matrix of the parameter estimates is then given by
$$R\left(\theta^{\mathrm{est}}\right)=B^{-1}\left(\theta^{\mathrm{est}}\right)\;C\left(\theta^{\mathrm{est}}\right)\;B^{-1}\left(\theta^{\mathrm{est}}\right).$$

Ideally, the off-diagonal elements of $R(\theta^{\mathrm{est}})$ should be as low as possible but several authors mention that 0.95 is a cut-off value above which the parameters cannot be estimated uniquely. In our context, the element $R_{1,2}$ of $R(\theta^{\mathrm{est}})$ holds special significance as it measures the interdependence between parameters *k* and η, and it is necessary for $R_{1,2}\in \left[-0.95,0.95\right]$ to ensure the independence of the parameters from each other.

The 100(1−α)% confidence level intervals for the parameters are as follows [36]:$${\left(\theta -\theta^{\mathrm{est}}\right)}^{⊺}\;S^{-1}\left(\theta^{\mathrm{est}}\right)\;\left(\theta -\theta^{\mathrm{est}}\right)\leq n_{\theta }\;s^{2}\;F\left(\alpha ,n_{\theta },n_{o}+1-n_{\theta }\right),$$
where $F(\alpha ,n_{\theta },n_{o}+1-n_{\theta })$ is the value of the *F* distribution with $\left(n_{\theta },n_{o}+1-n_{\theta }\right)$ degrees of freedom at a 1−α significance level and *s* is an approximation of the standard error of the prediction.

The visual analysis of the 95% confidence ellipsoids for pairs of parameters allows the identification of problems related to the capability of estimating the parameters uniquely. In practice, its orientation, which is a function of the elements of the parametric covariance matrix, is commonly used to diagnose parameter collinearity and local model identifiability issues [37].

## 4. Materials and Experimental Method

In this section, we characterize the material used in the sedimentation experiments and the experimental methodology.

All the experiments were carried out with calcium carbonate (CaCO_3_) particles. In the experiments, we used commercial calcium carbonate of analytical grade (full specification: *Calcium carbonate agr, reag.ph.eur.*, Labbox Export, Barcelona, Spain). The product has MW = 100.09 Da, bulk density equal to 0.760 kg/L, and meets the requirements for identifications A and B. The microscopy-based image of the particles is shown in Figure 3a. We note that the particles have irregular shapes. This effect is accounted for in the sedimentation model by parameter *k*. Figure 3b presents the particle size distribution (PSD) obtained by laser diffraction, using a *Mastersizer 3000* (Malvern Instruments Co., Ltd., Malvern, UK) granulometer. The average particle diameter is 4.51 μm. The density of the calcium carbonate was measured by helium pycnometry in a *MicroMeritics AccuPyc 1330 Pycnometer* (Micromeritics, Norcross, GA, USA). The value obtained was $\rho_{p}=$ 2.5327 g/cm^3^.

The water used in the experiments was previously distilled and was carried out at nearly 25 ${}^{\circ }C$. At this temperature, the density and viscosity of the water were $\rho_{l}=0.99705\;g/cm^{3}$ and $\mu_{l}=8.891\times {l0}^{-4}\;Pa\;s$, respectively. For these conditions, $v_{\mathrm{St}}=1.91\times {l0}^{-5}\;m/s$.

Previous to the experiments, a homogenized suspension with a given concentration, $c_{0}$, in CaCO_3_, was produced. Then, a graduated cylinder was filled up to a given height with the suspension. There, the suspension was homogenized and the initial height, $h_{0}$, was recorded. Then, the height of the interface was monitored throughout the experiment and its value was measured and recorded at each sampling time. During the initial stages, the frequency of the measurement is 1 min^−1^, and in the later stages, when its variation is small, the frequency decreases to a lower value. The resolution of the scale used to measure the height is 1 mm and the internal diameter of the cylinder is 4.5 cm. For convenience, in the remaining sections, we adopt derived units of the I.S. to represent the height of the interface and the velocity of the interface. The former is quantified in cm (of the cylinder) and the second in cm/min.

## 5. Numerical Results

In this section, we present the results obtained for a sequence of experiments, where the concentration of CaCO_3_ in the suspension varies. The values tested are 15 g/L, 20 g/L, 25 g/L, 30 g/L, and 35 g/L.

Figure 4 compares the data obtained for c(0)=25g/L and the respective model predictions when the velocity of settling and the height of the interface are fitted. Figure 4a compares the estimated settling velocity obtained using the numerical differentiation technique introduced in Section 3.1 with the model predictions obtained by solving the problem (11). Figure 4b compares the experimental data obtained for the height of the interface and the model predictions obtained by solving the problem (12). In both cases, we present the data, the model predictions, and the respective 95% confidence limits for the prediction. We note that the estimates based on $v_{t}\left(t\right)$ are less accurate than those based on h(t), as a result of the numerical differentiation introduced in Step 2 of the Algorithm. Consequently, the values of the settling velocity are somewhat dispersed and the 95% confidence limits appear relatively large. Contrarily, the 95% confidence limits for the height of the interface are remarkably tight, which is a consequence of the small error in the estimates. Figure 4b demonstrates that the model can adequately capture the main features of the system behavior.

Now, we compare the data with the model estimates for all solid concentrations tested. Figure 5a compares the predictions for the velocity of settling with data obtained via numerical differentiation and Figure 5b is for the height of the interface. The designations *est* and *exp* in the plots are for “estimated” and “experimental data”, respectively. The model prediction in the differential method is worse than the prediction by the integral method, due to the uncertainty introduced in the numerical differentiation of data. This behavior indicates that the direct use of the differential form of the model (in the integral method) produces more accurate parameters, as expected. These parameters are obtained in the sense of maximum likelihood (free of bias). However, the successful application of the integral method depends on the availability of good initial parameter estimates to avoid local solutions and other convergence difficulties. Hence, the initial application of the differential method to provide reasonable estimates of the numerical derivatives seems essential for the reliable use of the integral method.

Table 1 presents the values of the parameters and their respective confidence intervals, obtained for all concentrations. Column 1 contains the solid concentrations of the suspensions tested. Columns 2 and 3 present the estimates for the parameters *k* and η, respectively, and their respective 95% confidence ranges (after the ± sign). Column 4 presents the quality of the fit described by $R_{\mathrm{adj}}^{2}$, column 5 is for the parametric cross-correlation between *k* and η, column 6 contains the values of $d_{\mathrm{agg}}$ and column 7 contains the values of the mass-length fractal dimension $D_{f}$. These two quantities are obtained by relating *k* and η with the non-dimensional numbers ${\left(d_{\mathrm{agg}}/d_{p}\right)}^{D_{f}-1}$ and ${\left(d_{\mathrm{agg}}/d_{p}\right)}^{3-D_{f}}$ (see Equation (13) below) introduced by Heath et al. [21] and Nieto et al. [22]:
(13a)$$\begin{array}{cc} d_{\mathrm{agg}} & =\sqrt{k\;\eta }\;d_{p} \end{array}$$
(13b)$$\begin{array}{cc} D_{f} & =\frac{\ln(k)}{\ln(\sqrt{k\;\eta })}+1 \end{array}$$

Table 1 shows that parameter *k* has a notorious decrease as the concentration of solids increases; this means that higher concentration suspensions produce more aggregates with average smaller sizes. Contrarily, the η values are almost independent of the initial concentration of solids. The agreement between the estimates and the data is remarkably high (with $R^{2}$ above 0.9980 for all the concentrations), which indicates the accuracy of the prediction model. The parametric cross-correlation is below 0.95 for all the concentrations, evidencing that the parameters are independent of each other, making the model locally identifiable. The average diameter of the aggregates is between 10 and 17 times larger than the equivalent dimension of the individual particles and tends to decrease as the concentration increases. Finally, $D_{f}$ is almost independent of the initial concentration of the suspension, as expected from a mass-length characteristic dimension when the same material is used. The values of $D_{f}$ are in good agreement with that obtained by Heath et al. [38] for calcite (the value indicated is 2.4).

Figure 6 compares the 95% confidence regions for η versus *k*. We note that although the ellipse orientation indicates a certain degree of collinearity between the parameters, they can still be uniquely estimated.

## 6. Conclusions

We considered the problem of modeling the batch sedimentation of calcium carbonate particles from laboratory tests in graduated glass cylinders. Various experiments at different concentrations were carried out to analyze the effect of the initial concentration on the hindered settling velocity. The height of the interface between the supernatant zone and the clarified zone was monitored throughout the experiments. We proposed a systematic approach, introduced in Section 3, to fit a model to batch sedimentation data. This tool combines the differential and integral methodologies also used in the analysis of chemical kinetic data. Its structure involves: (i) The numerical differentiation of the height of the interface with a Lagrange-based polynomial approximation, to find local estimates of the velocity of settling, see Section 3.1; (ii) The fitting of the velocity of settling using the least squares algorithm with an integral approach, in Section 3.2; and (iii) The construction of numerical and graphical indicators to evaluate the model accuracy and the quality of fit, see Section 3.3.

The tool is demonstrated for the case of small-sized calcium carbonate particles, where the model used to describe the dynamics of the velocity of settling combines theoretical fundamentals with empirical knowledge. The phenomenological component follows from the balance of forces interacting in the particle; the empirical knowledge is used to account for the interactions between particles where two distinct effects are considered (i) the concentration of particles; and (ii) the aggregation of particles. This resulted in a simple but flexible two-parameter model, which can be related to other hindered settling velocity models in the literature. The tool proposed herein is general enough to find application in the treatment of batch sedimentation data for different systems, such as sludge, colloids, and metal oxide sedimentation, eventually with different or more complex kinetic models.

The error associated with the model structure is minimized by using a physically consistent representation that relies on (i) first principles-based knowledge; and (ii) empirical knowledge developed for similar systems. The observational error is taken into account to set the architecture behind the proposed tool. Specifically, our algorithm avoids fitting velocity data obtained by numerical differentiation, which may increase the influence of the observational noise; contrarily, we use the velocity data to provide a good parameter estimate that is then used as the initial solution to fit the original data to a differential model. The quality of fit metrics show that (i) the observational error allows identifying both model parameters independently (see the correlation matrix of the parameters), and (ii) the metrics capture the main features underlying the data (see the low magnitude of the standard error of the prediction). Finally, the numerical error is minimized by using accurate robust algorithms combined with good parameter initial estimates to fit the differential model. 

## Figures and Tables

**Figure 1 materials-16-04822-f001:**
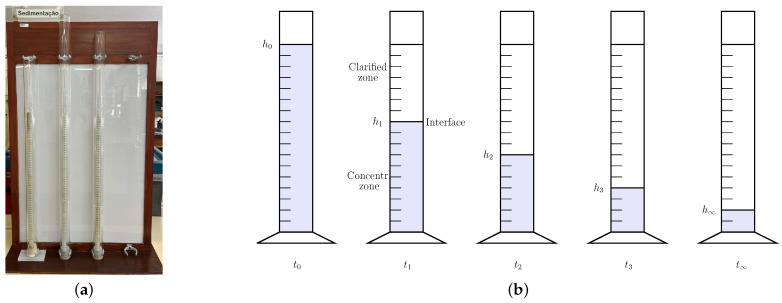
Representation of the batch sedimentation cylinders: (**a**) setup used in the experiments; (**b**) schematics of the dynamic behavior.

**Figure 2 materials-16-04822-f002:**
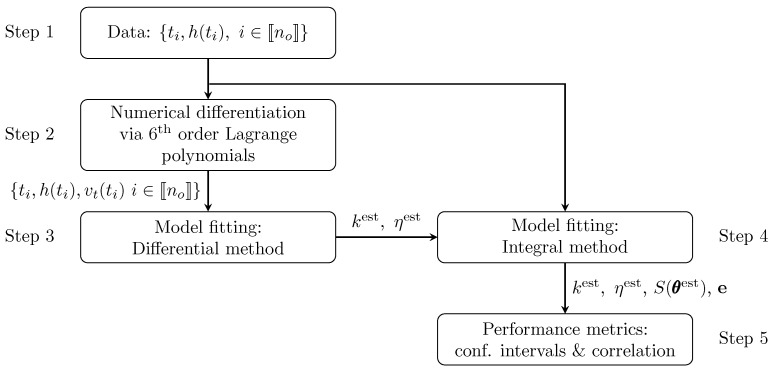
Structure of the systematic procedure used for model fitting.

**Figure 3 materials-16-04822-f003:**
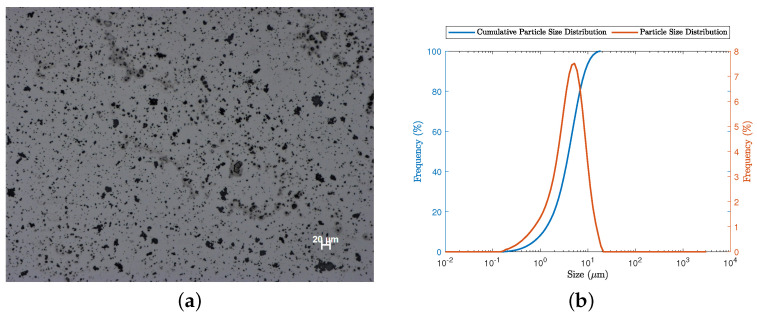
Characterization of the CaCO_3_ particles (**a**) concerning the shape, via microscopy analysis (graphic scale specified at the bottom-right of the picture; the horizontal bar corresponds to 20 μm length); (**b**) concerning the PSD, via laser diffraction with a Malvern *Mastersizer 3000*.

**Figure 4 materials-16-04822-f004:**
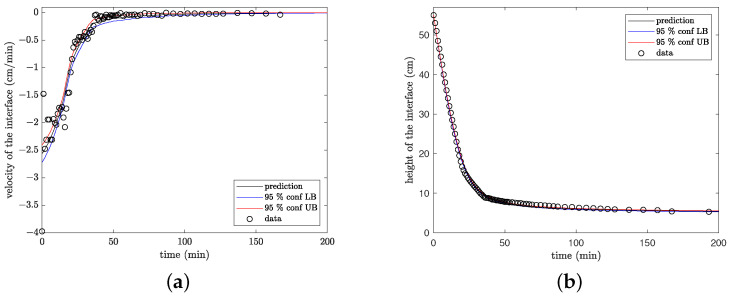
Experimental data and model predictions for the suspension with the concentration of solids of 25 g/L (**a**) based on the velocity of settling; (**b**) based on the height of the interface.

**Figure 5 materials-16-04822-f005:**
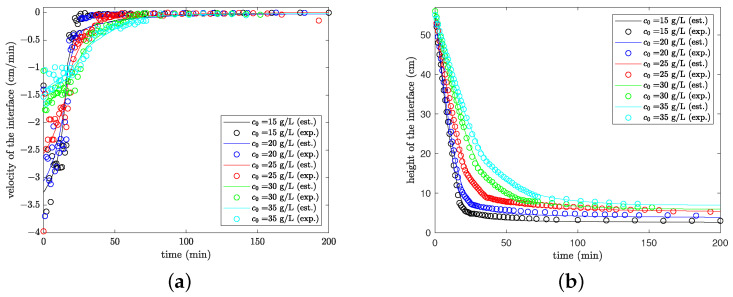
Experimental data and model predictions (**a**) based on the velocity of settling; (**b**) based on the height of the interface.

**Figure 6 materials-16-04822-f006:**
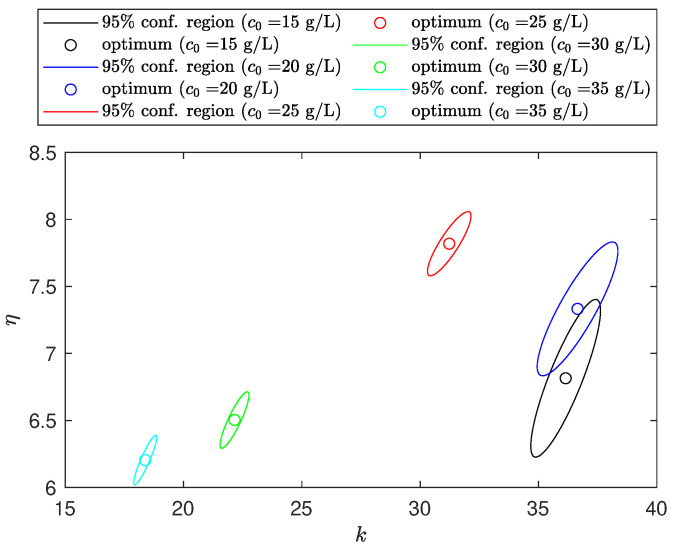
Parametric 95% confidence regions for all the concentrations.

**Table 1 materials-16-04822-t001:** Parameters obtained from model fitting with the algorithm of Figure 2.

Concentration (g/L)	$k^{\mathrm{est},\;†}$	$\eta^{\mathrm{est},\;†}$	$R_{\mathrm{adj}}^{2}$	$R_{1,2}$	$d_{\mathrm{agg}}$ (μm)	$D_{f}$
15	36.1462 ± 1.2081	6.8149 ± 0.4825	0.9983	0.8599	70.7845	2.3030
20	36.6484 ± 1.3957	7.3327 ± 0.4084	0.9981	0.8635	73.9326	2.2876
25	31.2257 ± 0.7476	7.8190 ± 0.1944	0.9991	0.8632	70.4707	2.2519
30	22.1557 ± 0.4993	6.5035 ± 0.1728	0.9996	0.8823	54.1369	2.2466
35	18.3790 ± 0.4001	6.2040 ± 0.1514	0.9998	0.9136	48.1586	2.2293

${}^{†}$ in format xx.xxxx±y.yyyy where xx.xxxx is the estimated value and y.yyyy is the 95% confidence interval width.

## Data Availability

The data can be provided on request.

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
