# Peer review of "Modeling the Batch Sedimentation of Calcium Carbonate Particles in Laboratory Experiments—A Systematic Approach"

_materials, 2023, doi:10.3390/ma16134822_

Round 1
Reviewer 1 Report
This paper designs a numerical model based on the experimental settling of calcium carbonate, which is instructive for the application of calcium carbonate settling and is recommended to be accepted with the following modifications:
1. the actual cylinder image of the sedimentation experiment should be added instead of the drawn one;
2. the amount of data in the PCA model used in this paper is too small, and it is suggested that the description in the literature of DOI: 10.1016/j.matdes.2023.111865.
have no ideas.
Author Response
Dear Reviewer,
We are sending you the revised version of the Manuscript materials-2446994 and our response
letter. In this revision we fixed the aspects identified by the reviewers, relative to the previous version of
the work.
In this revised version, we have tried our best to respond to each comment and suggestion made, and
therefore have introduced various changes, which are noted in colour.
Our point-to-point response follows immediately each paragraph of the original report. Overall, we are
grateful to the set of reviewers and the editorial team for a timely and careful review, which helped us to
significantly improve the quality of the manuscript.
Sincerely Yours,
Belmiro P.M. Duarte

Reviewer 2 Report
The present paper studied, by means of a mathematical model, the sedimentation by batches of calcium carbonate particles, based on Stokes' law and controlling the instantaneous velocity of sedimentation in order to adjust the sedimentation velocity model; and of course evaluate the adjustment by statistical methods The model to which it is adapted is biparametric and can be applied to sludges, colloids and metal oxides.
It is a very theoretical work with too much mathematical explanation that must be summarized, eliminating equations.
In addition, on lines 103, 110, 114, 116……..346, the equation number must be indicated in the text.
Line 183: Indicate that (7) is equation 7.
Line 289: Indicate the origin of the calcium carbonate: natural, synthetic, commercial, purity, chemical composition, mineralogical composition...?
Figure 3 a: Indicate size with graphic scale
Figures 4 and 5: the axis units do not belong to the I.S.
Author Response

(The authors gave the same response as above.)

Reviewer 3 Report
The paper entitled "Modeling the batch sedimentation of calcium carbonate particles in laboratory experiments — a systematic approach " by Maria J. Moura and co. addresses the numerical treatment of data obtained from batch sedimentation experiments of calcium carbonate particles.
The estimated and the experimental data fit well, however some revisions must be done, as follows:
1) Please explain more clearly what are the errors introduced by the model compared to the experimental data (in the paper the errors are only remembered )
2) Please place the figure 6 before the conclusions
Minor corrections must be done
Author Response

(The authors gave the same response as above.)

Round 2
Reviewer 1 Report
accepted
No comments
Reviewer 2 Report
Is OK